# Chemical Adjustment of Fibrinolysis

**DOI:** 10.3390/ph17010092

**Published:** 2024-01-10

**Authors:** Alexey M. Shibeko, Ivan S. Ilin, Nadezhda A. Podoplelova, Vladimir B. Sulimov, Mikhail A. Panteleev

**Affiliations:** 1Center for Theoretical Problems of Physicochemical Pharmacology, Russian Academy of Sciences, 109029 Moscow, Russia; alshibeko@gmail.com (A.M.S.); mapanteleev@yandex.ru (M.A.P.); 2National Medical Research Center of Pediatric Hematology, Oncology and Immunology Named after Dmitry Rogachev, 117197 Moscow, Russia; 3Research Computing Center, Lomonosov Moscow State University, 119991 Moscow, Russia; ilyin@dimonta.com (I.S.I.); vladimir.sulimov@gmail.com (V.B.S.); 4Dimonta, Ltd., 117186 Moscow, Russia

**Keywords:** thrombolysis, fibrin, plasmin, tissue plasminogen activator, ischemic stroke

## Abstract

Fibrinolysis is the process of the fibrin–platelet clot dissolution initiated after bleeding has been stopped. It is regulated by a cascade of proteolytic enzymes with plasmin at its core. In pathological cases, the balance of normal clot formation and dissolution is replaced by a too rapid lysis, leading to bleeding, or an insufficient one, leading to an increased thrombotic risk. The only approved therapy for emergency thrombus lysis in ischemic stroke is recombinant tissue plasminogen activator, though streptokinase or urokinase-type plasminogen activators could be used for other conditions. Low molecular weight compounds are of great interest for long-term correction of fibrinolysis dysfunctions. Their areas of application might go beyond the hematology field because the regulation of fibrinolysis could be important in many conditions, such as fibrosis. They enhance or weaken fibrinolysis without significant effects on other components of hemostasis. Here we will describe and discuss the main classes of these substances and their mechanisms of action. We will also explore avenues of research for the development of new drugs, with a focus on the use of computational models in this field.

## 1. Introduction

In response to vascular damage, the blood coagulation system forms a hemostatic fibrin–platelet plug that closes the site of injury and stops bleeding [1]. After the injury is repaired, the thrombus is dissolved in the course of the physiological processes of fibrinolysis. In addition to this crucial action, the combination of perpetual single platelet recruitment with fibrin formation and dissolution at the vessel wall is believed to be an important physiological process ensuring blood vessel homeostasis [2,3].

In pathological cases of thrombus formation disorders such as strokes or heart attacks, a thrombus can completely block the blood flow, causing ischemia of the surrounding tissues. In these cases, the thrombus can be removed with the help of special drugs that accelerate its dissolution, usually by promoting fibrinolysis [4]. In fibrinolytic system disorders, normal fibrin formation can be followed by an excessively rapid lysis (before the damaged vessel is repaired), leading to bleeding. On the contrary, fibrinolysis may be too slow, which leads to an increased risk of pathological thrombus formation. In addition, fibrinolysis may be an important player in processes that were not originally associated with it: for example, it is believed that bleeding in hemophilia is in part caused by a decreased resistance of hemophilic clots to fibrinolysis [5,6].

Stimulation of fibrinolysis is usually performed using physiological (urokinase plasminogen activator (UPA), tissue plasminogen activator (TPA)) or pathogen-derived (streptokinase) molecules. While some types of thrombosis could be treated by a wide variety of such molecules [7], the only licensed drug for emergency thrombus lysis in ischemic stroke is recombinant tissue plasminogen activator administered to a patient intravenously. However, when the fibrinolytic system malfunctions, orally available compounds with a low molecular weight are often used. These stimulate or weaken the blood clot-dissolving system with little if any effects on the other components of hemostasis.

## 2. Fibrinolysis

Blood coagulation is a cascade of enzymatic reactions initiated when blood comes into contact with the tissue under the endothelium lining the vessels. The blood coagulation network functions hand in hand with platelet-dependent hemostasis, mutually stimulating each other [8]. The main physiological agonists of hemostasis are collagen for platelets [9] and tissue factor, a glycoprotein that activates coagulation [10]. The blood coagulation cascade produces the serine protease thrombin. It activates fibrinogen to fibrin, which polymerizes and forms a plasma clot [11].

When a vessel wall is damaged, the main activator of clot lysis, called tissue plasminogen activator, is released from the tissues. In addition to this, another lysis activator, UPA, is constantly present in blood. Both these proteins are initially represented by a low-activity, single-chain form, which is proteolytically activated by plasmin to a more active, double-chain form. The main players of the fibrinolytic system, their sources, their targets, and their inhibitors are listed in Table 1.

The network of the TPA-induced fibrinolysis is shown in Figure 1. Lysis activators can convert plasminogen to plasmin directly in the solution, but this is a very slow reaction. Both plasminogen and TPA are able to bind fibrin; following this, the rate of plasminogen activation increases by orders of magnitude [12]. Plasmin bound to the fibrin network begins to cleave it, leading to a gradual dissolution of the clot.

In addition to the enzyme cascade, the lysis system contains plasmin inhibitors (antiplasmin, alpha-2 macroglobulin), plasminogen activator inhibitor PAI-1, and a special enzyme, TAFI (thrombin activatable fibrinolysis inhibitor).

Because of plasminogen activators in the circulation, fibrinolysis starts as soon as the fibrin becomes available. This occurs almost simultaneously with the initiation of blood coagulation [13]. However, it is slow unless additional activators are provided. The characteristic time for the appearance of a platelet plug is tens of seconds [1], while the fibrin mesh formation takes tens of minutes, and the dissolution of fibrin requires tens of hours.

One of the main regulators of fibrinolysis is thrombin, a protein that does not directly participate in the clot dissolution reactions. On the one hand, thrombin stimulates the release of TPA from the endothelium in a dose-dependent manner [14], thereby enhancing the lysis-activating signal. On the other hand, high thrombin concentrations result in a denser clot that is resistant to lysis [15]. Thrombin also activates factor XIII, which cross-links the fibrin clot, making it difficult to dissolve. In addition, thrombin in complex with thrombomodulin stimulates thrombin activatable fibrinolysis inhibitor (TAFI). This carboxypeptidase enzyme cleaves C-terminal lysines of the fibrin molecules used by plasmin(ogen) and TPA to bind to its surface, thereby also reducing the efficiency of fibrinolysis [16]. Thrombin also contributes greatly to procoagulant platelet formation, which impacts both coagulation and fibrinolysis [8,17,18].

## 3. Fibrinolytic System Pathologies and Targeting

### 3.1. Bleeding

Pathophysiological conditions of hyperfibrinolysis with an increased risk of bleeding disorders could arise from either congenital deficiency of one of the main inhibitors (α2-AP and PAI-1), or from an excess of TPA or UPA. However, the epidemiological frequency of these conditions is very low [19]. Much more frequently, hyperfibrinolysis is induced by the interaction of blood with foreign surfaces of the extracorporeal circuit, which was previously identified as an important factor of postoperative bleeding [20]. For this reason, intraoperative pharmacological treatment with antifibrinolytic drugs such as tranexamic acid (TXA) [21] or aminocaproic acid (EACA) is used to limit coagulation impairment.

Plasminogen circulates in a single-chain form, which consists of a plasminogen/apple/nematode (PAN) domain, five kringle domains, and a serine protease domain [22,23]. Plasminogen binds to the lysine residues on fibrin via an aminohexyl binding site in the K5 domain. The high-affinity lysine-binding sites in the K1, K2 and K4 domains interact with C-terminal lysines on the surface of fibrin, making the binding of plasminogen to fibrin more efficient [24]. The kringle domain ligands inhibit plasminogen binding to fibrin and cellular receptors and, therefore, suppress fibrinolysis. This is the basis of the antihemorrhagic action of lysine analogs such as tranexamic acid and aminocaproic acid [25]. The interaction of tranexamic acid and plasminogen is shown in detail in Figure 2.

The overall efficacy of TXA seems higher than that of EACA, as patients treated with EACA showed a significantly higher postoperative bleeding [26]. Both compounds have side effects: postoperative seizures occurred significantly more frequently in TXA patients, whereas EACA patients had a higher incidence of postoperative renal dysfunction [27]. Nevertheless, TXA is successfully used to control postoperative bleedings in many cases, including in arthroplasty [28], liver transplantations [29], cardiac surgery [30], general surgeries [31], and as a part of bleeding management in platelet function disorders [32]. It reduced mortality by 8% in non-surgical patients who were at risk for venous or arterial thrombosis [33]. However, TXA showed no or little efficacy in the treatment of traumatic brain injury [34] or cerebral hemorrhage [35]. On the one hand, this could simply be due to the insufficient importance of fibrinolysis inhibition in these pathologies. However, it could be alternatively hypothesized that this lack of effect could be related to sex-dependent effects of TXA on the BBB recently found in a murine model. It promoted BBB breakdown in the female mice and reduced BBB breakdown in the male mice. The supposed mechanism of this difference was an increased urokinase-type plasminogen activator level in the male mice [36], as TXA induces a conformational change in plasminogen that makes it more susceptible to proteolytic cleavage by UPA [37,38].

### 3.2. Thromboses

Studies of blood coagulation following elective surgery identified the state of fibrinolysis inhibition. In one study, it was associated with orthopedic surgery resulting in an increased risk of deep venous thrombosis and an increased risk of sepsis-provoked multiple organ failure [39]. In another, the state of the fibrinolysis inhibition was observed in approximately two-thirds of trauma patients [40].

Sepsis is often accompanied with disseminated intravascular coagulation (DIC) [41], systemic clot formation in the microcirculation of different organs [42]. This is caused by a combination of blood coagulation activation and decreased fibrinolysis. Sepsis leads to systemic endothelial activation, caused by inflammatory cytokines and hypoxia. This can increase TPA and PAI-1 antigen level [43], though changes in the TPA activity could be poorly detectable [44], most likely due to PAI-1-mediated inhibition. PAI-1 under normal conditions is downregulated by activated protein C [45]. However, protein C level is decreased in sepsis due to consumption [46], thus increasing PAI-1-induced inhibition of fibrinolysis.

A hypofibrinolytic state was observed in critically ill coronavirus disease 2019 (COVID-19) patients with high risk of thrombosis [47]. Patients with deep vein thrombosis demonstrated either low levels of TPA or elevated levels of PAI-1 [48,49]. Yet it was shown that patients with coronary artery stenosis had elevated levels of both PAI-1 and TPA [50], and a similar picture was observed for COVID-19 patients [51]. None of these fibrinolytic markers were associated with an increased risk of venous thromboembolism [52]. Thus, inhibition of PAI-1 can stabilize the coagulation state and improve the therapy outcome for some patients.

PAI-1 is secreted as an active protein that can form covalent complexes with UPA and TPA, thereby inhibiting both plasminogen activators. Active PAI-1 is not stable and is spontaneously converted to a stable non-reactive (latent) conformation with an apparent half-life of 1–2 h under normal physiological conditions. This non-reactive form does not interact with its target proteases [53]. PAI-1 inhibits TPA and UPA very rapidly with second-order rate constants of about 10^7^ M^−1^ s^−1^. The key mechanism of this reaction is that plasminogen activator recognizes PAI-1 as a (pseudo)substrate. It is generally accepted that the P1 residue is the major determinant of the protease specificity of serpins [54].

PAI-1 level is increased in septic patients and other cases of prothrombotic situations, e.g., ischemic cardiovascular disease [55]. This may be the reason for the fibrinolytic system downregulation, which shifts the delicate balance of clot formation and dissolution to the procoagulant state. Therefore, inhibition of this molecule can be beneficial.

A large virtual screen of a library of commercially available chemicals [20] followed by docking on PAI-1 with a focus on the cleft in β-sheet A led to the discovery of TM5007 [56]. It is a small molecule which prevents PAI-1/PA complex formation [57]. It was shown to be effective as an antithrombotic agent (which does not prolong bleeding time, PT, or APTT) in a arteriovenous shunt murine model and ferric chloride-induced thrombosis murine model [56].

Further studies of the TM5007 activity and structure led to the development of its derivative TMM527, a low molecular weight compound with an improved inhibitory profile and better oral bioavailability [58]. It binds strand 4 of the A β-sheet (s4A) at the P14-P9 position of PAI-1. The interaction of TM5275 and PAI-1 is shown in Figure 3.

TM5275 showed antithrombotic efficiency in rats and nonhuman primates [59]. TM5275 did not affect either activated partial thromboplastin time or the prothrombin time, and did not prolong bleeding. Combined administration of TM5275 and TPA improved the therapeutic efficacy of the latter without additional adverse effects compared with TPA alone. TM5275 demonstrated decreased liver fibrosis in different rat metabolic syndrome models [60] and effectively inhibited albuminuria, mesangial expansion, extracellular matrix accumulation, and macrophage infiltration in diabetic kidneys [61]. However, no human trials with this compound have been performed yet.

Further optimization of the TM5007 and TM5275 structures combining substituted anthranilic acid and lipophilic moieties with the proper length of acyl-type linker led to the discovery of the novel compounds TM5441 and TM5484 [62]. TM5441 increased collateral perfusion and reduced infarction in spontaneously hypertensive rats subjected to ischemia by means of middle cerebral artery occlusion [63]. It also prevented body weight gain and systemic insulin resistance induced by a high-fat diet in C57BL/6J mice [64]. TM5481 ameliorated paralysis, attenuated demyelination, and affected spinal cord axonal degeneration in the murine model of allergic encephalomyelitis [65].

## 4. In Silico Approaches to Identifying New Targets in the Fibrinolytic System

Correction of fibrinolytic system disorders or tweaking it in order to rebalance the hemostatic system is a strategic challenge in the field. Thrombolytic therapy, e.g., the one used for ischemic stroke patients, cannot be considered a correction, as plasminogen activators are administrated in concentrations hundreds of times higher than the physiological ones. This is required for the urgent removal of a life-threatening thrombus, but intracranial bleeding is a major complication of this approach. However, a more sophisticated tool is needed to fine-tune the system and find the right balance between thrombosis and bleeding. In order to develop such tools, one has to determine the drug targets, or rather the control mechanisms, best suited to become the focus of the therapy. A powerful approach for this could be simulations in computational systems biology [66,67] and quantitative systems pharmacology [68].

Use of computer models allows analysis of the system sensitivities for all parameters even when the system is complex, spatially heterogeneous, and includes various processes of a diverse nature, such as biochemical reactions, mass transport due to diffusion and advection, and mechanical or electrical ones [69]. This could be combined with advanced tools such as analysis of the timescale hierarchy [70,71]. It could ultimately be possible to propose a modular decomposition of a system with the identification of subsystems responsible for specific tasks [72].

For example, a detailed analysis of the sensitivity of thrombolysis in a spatially non-uniform system showed that the penetration of TPA into the clot during thrombolytic therapy played a great role in its efficacy [73]. To achieve this, a PDE model of fibrin clot lysis was coupled with the equations of Darcy’s law describing liquid permeation through a porous medium of the clot. The simulations were performed in a 1D area, where the left half of the domain was occupied by the fibrin clot, whereas the right half of the domain included blood with TPA. On the right boundary, Dirichlet conditions for the transportable reactants TPA, plasminogen, and PAI-1 were set. These boundary concentrations were equal to the initial condition because of the steady supply. Impressively, PAI-1 level regulated the dynamics of clot lysis, although the plasma PAI-1 concentration was substantially below that of therapeutic TPA concentrations. The efficacy of lysis decreased with the increase of the PAI-1 concentration and/or the flow-induced advection of TPA caused by the increase of the pressure difference (Figure 4). The mechanism of the unexpected control of lysis by PAI-1 was related to the spatial heterogeneity itself: PAI-1 concentration and TPA concentration at the boundary of the clot-penetrating front were comparable. Accordingly, the PAI-1 level determined the recanalization time under flow conditions in collateral vessels. Based on this, it can be assumed that the combination of the TPA therapy and a PAI-1 inhibitor similar to TM5275 could be significantly more effective than just TPA without worsening the adverse effects, which was shown in vivo [59]. A similar effect could be achieved by using a PAI-1-resistant variant of TPA [74], but the combination of an already developed TPA-based therapy with a small molecule could be more cost-effective.

The same analysis showed that specific recognition and inhibition of the fibrin-bound plasmin can be particularly efficient to downregulate clot lysis. The fibrin-bound form of the enzyme is an attractive target because the half-life time of free plasmin is very short, but fibrin produces a major protecting effect. Although the overall plasmin-inhibiting ability of plasma may seem exceptional, this ability is not really implemented because of this heterogeneity. If a small molecule able to specifically interfere with bound plasmin is developed, it may significantly correct hyperfibrinolysis without affecting the other effects of the plasmin system not related to blood coagulation.

As another example, different types of computational simulation approaches were used to identify several PAI-1 inhibitors. For example, a phenolic component of olive oil oleuropein was identified as a potent PAI-1-binding molecule with the help of molecular docking analysis. It is likely to act as a natural PAI-1 inhibitor by incrementally and selectively affecting PAI-1 levels in ER-/PR- breast cancer cells [75]. A combination of molecular docking, molecular dynamics, and in vitro chromogenic assays was used to identify the Dracaena dragon blood tree chemical constituents directly targeting PAI-1; this also led to the discovery of a compound that does not bind to the active conformation of PAI-1, but rather binds to and stabilizes the latent form [76].

In silico modeling of fibrinolysis was previously used to evaluate the effect and action mechanisms of different thrombolytic agents in acute ischemic stroke [77]. A pharmacokinetic–pharmacodynamic model of TPA with local clot lysis dynamics including plasminogen, plasmin, fibrinogen, PAI-1, antiplasmin, and a2-macroglobulin, as well as a permeable fibrin clot, was adopted from [78]. It demonstrated that urokinase was highly efficient in complete clot lysis, but is predicted to have the highest risk of intracranial bleeding due to excessive fibrinogen depletion. Tenecteplase and alteplase had comparable thrombolysis efficacies, but tenecteplase has a lower risk of ICH and better resistance to PAI-1. Reteplase had the lowest fibrinolysis rate, but fibrinogen concentration in systemic circulation was unaffected during thrombolysis.

Thus, to successfully correct the fibrinolytic system, our knowledge of its structure and functioning has to be improved. Both chemical reactions and the physical conditions of their occurrence (spatial distribution of the reactions, the presence of blood flow carrying the substances in space, etc.) require elucidation. This brings us to the main limitations of the computational approach in studying biological processes, such as uncertainty in concentration distribution of the participating species. For example, plasminogen activators and inhibitors are elevated in the vicinity of the thrombus, and their average plasma level may not reflect the correct local concentrations. This limits our understanding of the transport processes, which are crucial for the correct spatial organization of fibrin clot lysis.

The aforementioned difficulties can be overcome in future by means of improved experimental and computer simulation techniques. One can use in silico modeling to screen a wide range of species concentrations and distribution as well as mechanisms of action and activities and propose experiments for validation. A detailed analysis of sensitivity and necessity [70] can provide information on the regulatory mechanisms and potential molecular targets. Molecular dynamics and docking could then be used to identify the most prominent compounds for synthesis and experimental evaluation.

## Figures and Tables

**Figure 1 pharmaceuticals-17-00092-f001:**
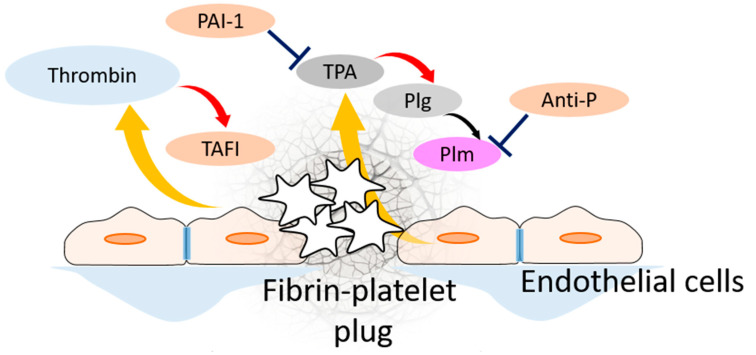
Basic sketch of the fibrinolytic system. Rupture of endothelial wall causes blood clotting and fibrin–platelet plug seals the leakage. Thrombin produced during this process activates TAFI, which block the ability of TPA and plasminogen (Plg) to bind fibrin. TPA emerges from endothelial cells in response to thrombin. Upon binding fibrin, it activates fibrin–plasminogen. Upon activation, plasmin (Plm) cleaves and destroys fibrin mesh. PAI-1 inhibits TPA, while antiplasmin (Anti-P) inhibits plasmin, both free and fibrin-bound.

**Figure 2 pharmaceuticals-17-00092-f002:**
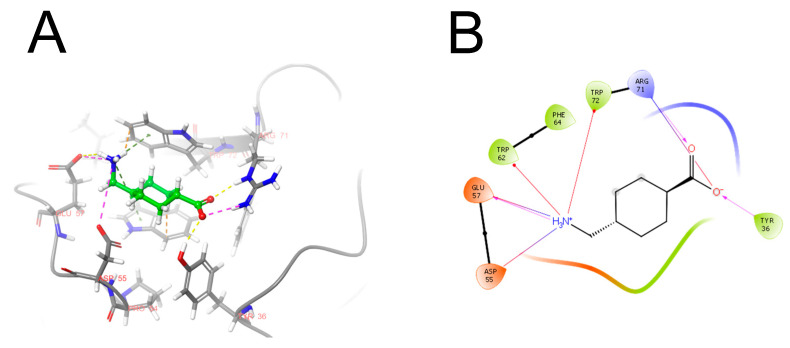
The structure of tranexamic acid in a complex with plasminogen (PDB ID: 1B2I protein structure prepared using Protein Preparation Wizard, Schrodinger package): (**A**) pose in the plasminogen. Carbon atoms of protein are painted gray, and ligand’s carbon atoms are painted green. Yellow dashed lines indicate hydrogen bonds, green indicates pi-cationic interaction, and violet indicates electrostatic interaction; (**B**) 2D diagram of interactions. Tranexamic acid forms several specific interactions when it binds to the lysine recognition site of plasminogen. The carboxyl group forms a salt bridge with the ARG71 residue, as well as a hydrogen bond with the TYR36 hydroxyl group. The positively charged amino group enters into a pi-cationic interaction with two tryptophan residues: TRP62 and TRP72. In addition, this group is located close to the GLU57 residue, forming a hydrogen bond with its carboxyl group. Violet arrows indicate hydrogen bonds, yellow indicate halogen interaction, red indicate pi-cationic interaction. The drawing was prepared in the Maestro program, Schrödinger, LLC. Version 5.1.139, Release 2020-3.

**Figure 3 pharmaceuticals-17-00092-f003:**
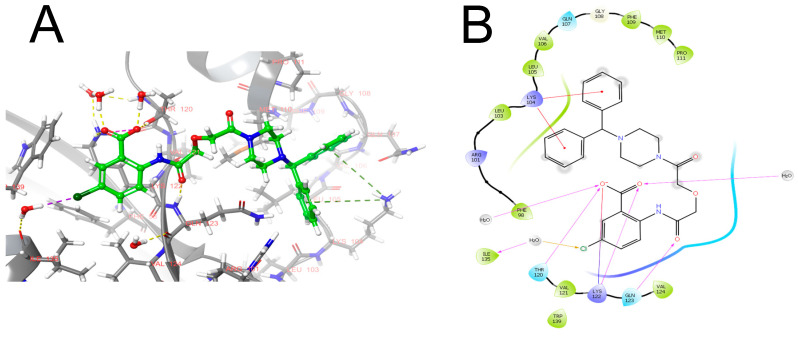
Predicted position of TM5275 in complex with PAI-1 (protein structure is taken from 7AQF; docking was performed in the Glide program): (**A**) pose in the protein. Carbon atoms of protein are painted gray, and ligand’s carbon atoms are painted green. Yellow dashed lines indicate hydrogen bonds, green indicates pi-cationic interaction and violet indicates electrostatic interaction. (**B**) 2D diagram of interactions. The carboxychlorophenyl fragment of TM5275 binds near LYS122, forming two hydrogen bonds with the charged amino group of this residue: one through a direct H-bond, and the other through a water molecule. In addition, the chlorine atom enters into a halogen bond through the rest of the water with ILE135. The central part of TM5275 forms a hydrogen bond with the nitrogen atom of the backbone belonging to GLN123. The terminal diphenyl fragment binds in a small hydrophobic cavity and forms a pi-cationic interaction with LYS104. Violet arrows indicate hydrogen bonds, yellow indicate halogen bonds, red indicate pi-cationic interaction. The drawing was prepared in the Maestro program Schrödinger, LLC. Version 5.1.139, Release 2020-3.

**Figure 4 pharmaceuticals-17-00092-f004:**
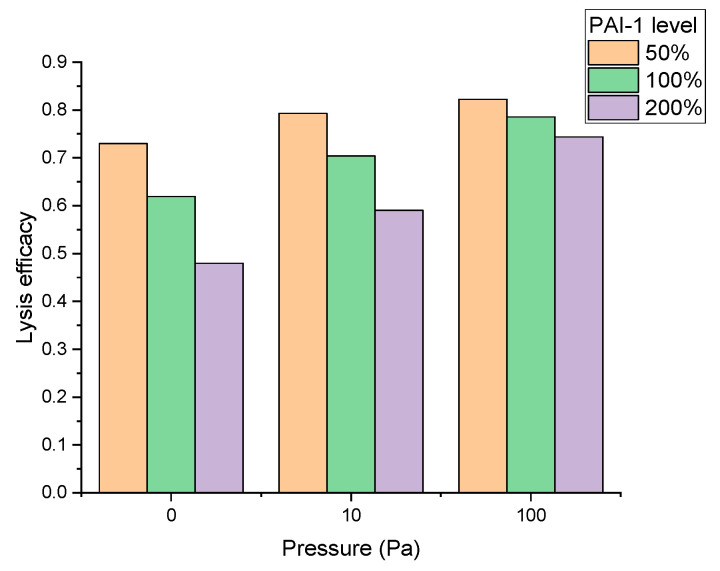
The effect of PAI-1 on the spatial clot lysis by TPA in the computer model depends on physical conditions. The efficacy of lysis (the area of the dissolved part of the clot) was simulated for different PAI-1 concentrations (50%, 100%, and 200% of the mean plasma level). Simulations were carried out for either no flow conditions or for blood flow produced by a pressure difference of 10 Pa or 100 Pa. The TPA concentration was 30 nM, while the PAI-1 mean plasma level was assumed to be 2 nM. Based on the data from [73].

**Table 1 pharmaceuticals-17-00092-t001:** The main participants of the fibrinolytic system.

Participant of Fibrinolytic System	Its Source	It Is Activated by	Its Target	It Is Inhibited by
TPA	Endothelial cells	plasmin	Activates plasminogen	PAI-1
UPA	Present in blood plasma		Activates plasminogen	PAI-1
PAI-1	Platelets, present in blood plasma		Inhibits TPA and UPA	Activated protein C, autoinactivation
Plasmin(ogen)	Present in blood plasma	UPA, TPA, TPA bound with fibrin	Cleaves fibrin	Antiplasmin, a2-macroglobulin
Fibrin(ogen)	Present in blood plasma	Thrombin		
TAFI	Present in blood plasma	Thrombin–thrombomodulin complex	Cleaves C-terminal lysines on fibrin molecules, which prevents TPA and plasmin(ogen) binding	autoinactivation

## Data Availability

Data sharing not applicable.

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
