# Peer review of "Chemical Adjustment of Fibrinolysis"

_pharmaceuticals, 2024, doi:10.3390/ph17010092_

Round 1

Reviewer 1 Report

Comments and Suggestions for Authors

Overall, the authors have provided a relatively short  narrative review about the pathophysiology behind fibrinolyting derangements, and the main drugs currently used for the management of these disorders. English editing is required as well.

Comments:

11. Please rephrase the following sentences so the meaning is more clear. There are some grammar errors as well.

“Stimulation of fibrinolysis is usually performed using enzymes of high molecular  weight enzyme activators, inspired by physiological (urokinase plasminogen activator  (UPA), tissue plasminogen activator (TPA))s or pathogen-derived (streptokinase) molecules”.

“Another, much more frequent complication of hyperfibrinolysis is its manifestation induced by the interaction of blood with foreign surfaces of the extracorporeal circuit, which have been identified by previously as important factors of postoperative bleeding”

22. In section 3.2

The derangements of fibrinolysis in the setting of sepsis should be also mentioned, as described in several studies i.e. Tsantes AG, Parastatidou S, Tsantes EA, et al. Sepsis-Induced Coagulopathy: An Update on Pathophysiology, Biomarkers, and Current Guidelines. Life (Basel). 2023 Jan 28;13(2):350. doi: 10.3390/life13020350. PMID: 36836706; PMCID: PMC9961497.

Comments on the Quality of English Language

Extensive editing is required

Author Response

Reviewer #1

Overall, the authors have provided a relatively short  narrative review about the pathophysiology behind fibrinolyting derangements, and the main drugs currently used for the management of these disorders. English editing is required as well.

Comments:

Please rephrase the following sentences so the meaning is more clear. There are some grammar errors as well.

“Stimulation of fibrinolysis is usually performed using enzymes of high molecular  weight enzyme activators, inspired by physiological (urokinase plasminogen activator  (UPA), tissue plasminogen activator (TPA))s or pathogen-derived (streptokinase) molecules”.

Thank you, we have changed rephrased the text, and now it is: “Stimulation of fibrinolysis is usually performed using physiological (urokinase plasminogen activator (UPA), tissue plasminogen activator (TPA)) or pathogen-derived (streptokinase) molecules”.

“Another, much more frequent complication of hyperfibrinolysis is its manifestation induced by the interaction of blood with foreign surfaces of the extracorporeal circuit, which have been identified by previously as important factors of postoperative bleeding”

Thank you, we have changed rephrased the text, and now it is: “Much more frequently hyperfibrinolysis is induced by the interaction of blood with foreign surfaces of the extracorporeal circuit, which have been previously identified as important factors of postoperative bleeding”.

In section 3.2

The derangements of fibrinolysis in the setting of sepsis should be also mentioned, as described in several studies i.e. Tsantes AG, Parastatidou S, Tsantes EA, et al. Sepsis-Induced Coagulopathy: An Update on Pathophysiology, Biomarkers, and Current Guidelines. Life (Basel). 2023 Jan 28;13(2):350. doi: 10.3390/life13020350. PMID: 36836706; PMCID: PMC9961497.

Thank you, we have added the following text: “Sepsis, especially at its last and severe stage, septic shock, is often accompanied with disseminated intravascular coagulation (DIC) (doi.org/10.1186/s40560-016-0149-0), a condition that causes widespread thrombosis in the microcirculation of different organs (10.1016/j.cccn.2004.02.015). This is caused by combination of increased blood coagulation and decreased fibrinolysis. Systemic endothelial activation, caused by inflammatory cytokines and hypoxia during sepsis, increase TPA and PAI-1 antigen level (10.1016/j.chest.2017.01.010), but TPA activity is usually undetectable (10.1111/j.1365-2141.1990.tb02623.x), most likely due to PAI-1 mediated inhibition. PAI-1 under normal conditions can be inhibited by activated protein C (10.1055/s-0038-1651089), but in sepsis protein C level decline due to consumption (10.1378/chest.101.3.816), thus increasing PAI-1 induced inhibition of fibrinolysis.”.

Extensive English language editing is required

Thank you, we have worked through the text and double-checked the grammar.

Reviewer 2 Report

Comments and Suggestions for Authors

Comments to the Authors:

The manuscript makes an ambitious attempt to present fibrinolysis; however, it requires minor revisions to enhance clarity and overall understanding.

 Fig.1: The figure needs simplification for better clarity and self-explanation. Currently, it appears congested and is challenging to interpret. Consider modifying it to ensure basic understanding for the readers.

Page 3, Section 3: Fibrinolytic System: The text lacks clarity regarding the players or targets of the fibrinolytic system. It would be beneficial to provide tabulated information or a flowchart, specifically detailing plasminogen activators, plasminogen activator inhibitors, and plasmin inhibitors.

Fig.4 Methodology: The manuscript lacks details on the methodology for Fig.4. It is crucial to provide relevant information within the manuscript to enhance transparency.

Limitations and Scope: A few sentences outlining the limitations and scope of the study should be included. Additionally, the conclusions drawn from the study need to be explicitly stated.

The manuscript holds promise but requires these minor revisions to ensure clarity and completeness. I recommend acceptance after addressing these comments.

Author Response

Reviewer #2

The manuscript makes an ambitious attempt to present fibrinolysis; however, it requires minor revisions to enhance clarity and overall understanding.

 Fig.1: The figure needs simplification for better clarity and self-explanation. Currently, it appears congested and is challenging to interpret. Consider modifying it to ensure basic understanding for the readers.

Thank you, we have updated the figure to make it more straightforward for understanding.

Page 3, Section 3: Fibrinolytic System: The text lacks clarity regarding the players or targets of the fibrinolytic system. It would be beneficial to provide tabulated information or a flowchart, specifically detailing plasminogen activators, plasminogen activator inhibitors, and plasmin inhibitors.

Thank you, we have added the table 1 with the main players of the fibrinolytic system, their source, their targets and inhibitors.

Fig.4 Methodology: The manuscript lacks details on the methodology for Fig.4. It is crucial to provide relevant information within the manuscript to enhance transparency.

Thank you, we have added methodology of the work, results of which are presented in the fig.4: “Briefly, in this study a PDE model of fibrin clot lysis was coupled with the equations of the Darcy’s law describing liquid permeation through a porous medium of the clot. Simulations were performed in a 1-d area, left half of the domain is occupied by the fibrin clot, whereas the blood with TPA is in the right half of the domain. On the right boundary, Dirichlet conditions for the transportable reagents TPA, plasminogen, and PAI-1 were set. Their boundary concentrations were equal to the initial condition be-cause of constant refreshment of all fibrinolysis factors and continuous infusion of TPA.”.

Limitations and Scope: A few sentences outlining the limitations and scope of the study should be included. Additionally, the conclusions drawn from the study need to be explicitly stated.

Thank you, we have added more information about limitations of the study and conclusions. Now, the text is the following: “Thus, for a successful achievement of the goal of controlled correction of fibrinolytic system, we need to know its structure and functioning, both chemical reactions and the physical conditions of their occurrence (spatial distribution of reactions, the presence of a flow (blood flow) carrying the substances in space, etc. It brings us to the main limitation of the computational approach in studying biological processes, in particular, uncertainty in concentrations of species. Plasminogen activators and inhibitors are elevated in the vicinity of the thrombus formation area, and their average plasma level shift may not reflect the correct changes of their local concentrations. It worsens our understanding of the impact of transport processes, which are crucial due to localized sites of fibrinolysis and coagulation systems participants’ production and activation.

Yet, the aforementioned difficulties can be overcome to some extent by both more accurate measurements in vivo/in vitro, and by performing rigor simulations in order to describe the observed picture. Using in silico modeling, we can look through a number of cases with different species concentrations and distribution, different mechanisms of action and different activity, picking up the best fitting variant, which can be validated experimentally. A detailed analysis of sensitivity and specificity of reactions can provide us with information about regulatory mechanisms and potential molecular targets, and we can test them in silico, finding the most prominent variants for further synthesis and evaluation in vitro/ in vivo”.

The manuscript holds promise but requires these minor revisions to ensure clarity and completeness. I recommend acceptance after addressing these comments.

Reviewer 3 Report

Comments and Suggestions for Authors

The review manuscript by Shibeko and colleagues discussed blood coagulation and fibrinolysis in summary of previously published literature. References are reasonably well organized and cited. A few comments for authors to consider during manuscript revision:

It was unclear as whether the images presented in figures 1 to 3 were adapted from previous publications or were produced by authors of this manuscript as summarized graphic presentation of previous observations. If adapted from previous publications, authors should provide key references at the text and figure legend to credit the original authors. If current authors developed these images as a summary of previous observations, then multiple references should be cited at each figure and related text such that readers can track down the original observations. 

Discussions about TM5275 and TM5007 as specific PAI-1 inhibitors with antithrombotic effects should be an important part of this review manuscript. Images in Figure 3 illustrating TM5275/PAI-1 interaction are very good. More references should be provided of the original work concerning TM5275/TM5007 chemical structure design, their specific interactions with PAI-1, and their utility/efficacy in various disease models/clinical settings.  

Text in section 4 made good arguments about the usage of computational simulation in identifying new targets in fibrinolytic system. Discussion about TPA and PAI-1 are well presented. It would strengthen the review significantly if authors could provide a few good examples of new chemicals/small molecules suggested by using the computational simulation approach that could interact with PAI-1 with potential efficacy in disease models/clinical settings.  

When using abbreviations, define each term at its first appearance and then use the defined term consistently in the manuscript. Some of the terms were used without being defined first (such as Covid-19 and DVT, as this reviewer can see it).    

Comments on the Quality of English Language

Some English editing is recommended 

Author Response

Reviewer #3

The review manuscript by Shibeko and colleagues discussed blood coagulation and fibrinolysis in summary of previously published literature. References are reasonably well organized and cited. A few comments for authors to consider during manuscript revision:

It was unclear as whether the images presented in figures 1 to 3 were adapted from previous publications or were produced by authors of this manuscript as summarized graphic presentation of previous observations. If adapted from previous publications, authors should provide key references at the text and figure legend to credit the original authors. If current authors developed these images as a summary of previous observations, then multiple references should be cited at each figure and related text such that readers can track down the original observations.

Thank you for this question. Fig.1 was drawn by authors as a representation of the text in the section 2, where process of fibrinolysis is described with the corresponding references.

Fig.2 and Fig.3 were created by authors and illustrate the interactions of small molecules with plasminogen and PAI-1 respectively, based on the information about their crystal structure.

Discussions about TM5275 and TM5007 as specific PAI-1 inhibitors with antithrombotic effects should be an important part of this review manuscript. Images in Figure 3 illustrating TM5275/PAI-1 interaction are very good. More references should be provided of the original work concerning TM5275/TM5007 chemical structure design, their specific interactions with PAI-1, and their utility/efficacy in various disease models/clinical settings.

Thank you, we have updated the information on TM5007, TM5271 and added some data on their derivative compounds TM5441 and TM5484.

Text in section 4 made good arguments about the usage of computational simulation in identifying new targets in fibrinolytic system. Discussion about TPA and PAI-1 are well presented. It would strengthen the review significantly if authors could provide a few good examples of new chemicals/small molecules suggested by using the computational simulation approach that could interact with PAI-1 with potential efficacy in disease models/clinical settings.

Thank you about this commentary, we have added some information about such research: “Different types of computational simulation approaches were used to identify several potent PAI-1 inhibitors. For example, with the help of molecular docking analysis oleuropein, a phenolic component of olive oil, was identified as a potent PAI-1-binding molecule, and it may act as a natural PAI-1 inhibitor by incrementally destabilising PAI-1 levels selectively in ER-/PR- breast cancer cells (10.1007/s10549-020-06054-x). Combination of molecular docking, molecular dynamics and in vitro chromogenic assays was used to identify the Dracaena dragon blood tree chemical constituents that can directly target PAI-1, and found a compound that does not bind to the active conformation of PAI-1, but bind to and stabilize the latent form (10.1002/bab.2100).

In silico modeling of fibrinolysis was used to evaluate the effect of different thrombolytic agents in acute ischemic stroke (10.3390/pharmaceutics15030797). A pharmacokinetics-pharmacodynamics model of TPA with local clot lysis dynamics, including plasminogen, plasmin, fibrinogen, PAI-1, antiplasmin and a2-macroglobulin, as well as permeable fibrin clot was adopted from (10.3390/pharmaceutics11030111). It demonstrated that urokinase has the quickest lysis completion but the highest risk of intracranial bleeding due to excess fibrinogen depletion; tenecteplase and alteplase have very similar thrombolysis efficacy, tenecteplase has a lower risk of ICH and better resistance to PAI-1; reteplase has the slowest fibrinolysis rate, but fibrinogen concentration in systemic plasma is unaffected during thrombolysis”.

When using abbreviations, define each term at its first appearance and then use the defined term consistently in the manuscript. Some of the terms were used without being defined first (such as Covid-19 and DVT, as this reviewer can see it).    

Thank you, we have corrected it.

Comments on the Quality of English Language

Some English editing is recommended 

Thank you, we have worked through the text and double-checked the grammar.

Round 2

Reviewer 1 Report

Comments and Suggestions for Authors

There are still serious issues with the English laguage, and the manuscript is difficult to follow. Moreover, the phrases that I had difficulty to understand and I asked to be modified in the first version of the  manuscript, have not been improved in the revised manuscript. I beleive that this manuscript is not suitable for publication.  

Comments on the Quality of English Language

Needs editing

Author Response

Thank you very much for taking the time to review this manuscript. We carefully corrected the text and hope that the readability of the revised version is significantly improved. All essential modifications are marked with yellow background in the revised manuscript.

Round 3

Reviewer 1 Report

Comments and Suggestions for Authors

I still stand on my previous opinion, this manuscript is not suitable for publication.  

Comments on the Quality of English Language

Major english editing is needed